# Advancements in the Alcohol-Associated Liver Disease Model

**DOI:** 10.3390/biom12081035

**Published:** 2022-07-27

**Authors:** Lin Zhu, Hai-Di Li, Jie-Jie Xu, Juan-Juan Li, Miao Cheng, Xiao-Ming Meng, Cheng Huang, Jun Li

**Affiliations:** Inflammation and Immune Mediated Diseases Laboratory of Anhui Province, Anhui Institute of Innovative Drugs, School of Pharmacy, Anhui Medical University, Hefei 230032, China; zhulin20040629@163.com (L.Z.); lihaidi@hku.hk (H.-D.L.); jiejie141225@163.com (J.-J.X.); lijuanjuan202205@163.com (J.-J.L.); chengmiao0505@163.com (M.C.)

**Keywords:** alcohol, alcohol-associated liver disease, pathological processes, in vitro model, in vivo model

## Abstract

Alcohol-associated liver disease (ALD) is an intricate disease that results in a broad spectrum of liver damage. The presentation of ALD can include simple steatosis, steatohepatitis, liver fibrosis, cirrhosis, and even hepatocellular carcinoma (HCC). Effective prevention and treatment strategies are urgently required for ALD patients. In previous decades, numerous rodent models were established to investigate the mechanisms of alcohol-associated liver disease and explore therapeutic targets. This review provides a summary of the latest developments in rodent models, including those that involve EtOH administration, which will help us to understand the characteristics and causes of ALD at different stages. In addition, we discuss the pathogenesis of ALD and summarize the existing in vitro models. We analyse the pros and cons of these models and their translational relevance and summarize the insights that have been gained regarding the mechanisms of alcoholic liver injury.

## 1. Introduction

Alcohol-associated liver disease (ALD) is a major cause of chronic liver disease worldwide [1]. According to the degree of liver injury, ALD usually first manifests as steatosis and then develops into alcoholic hepatitis (AH) or even more severe fibrosis, cirrhosis, and superimposed hepatocellular carcinoma (HCC) [2]. Alcoholic fatty liver (AFL) is usually considered the initial pathological process. It is now recognized as a risk factor for the development of liver fibrosis and cirrhosis [3,4]. Fatty liver, fibrosis, and hepatitis can occur in the same patient separately, simultaneously, or successively [5]. In alcoholic steatohepatitis (ASH), steatosis is frequently accompanied by hepatocyte necrosis, apoptosis, and mild inflammation. In contrast to ASH, alcoholic hepatitis is characterized by severe necrosis, apoptosis, and the inflammation of liver cells in the absence of steatosis [6]. However, chronic alcoholic steatohepatitis and liver fibrosis usually coexist [7]. Among alcoholics, approximately 30–50% of AH patients will develop fibrosis and cirrhosis when there is severe ethanol-induced liver damage, which may lead to HCC in some cases, while patients without alcoholic hepatitis have a lower risk of cirrhosis [8,9,10].

Alcohol consumption and diet are the key triggers that affect the progress of ALD. Therefore, the application of animal models to mimic human drinking patterns is helpful for studying the pathogenesis and treatment strategies of ALD. Animal models that are currently in use include the Lieber–DeCarli diet feeding model, the Tsukamoto–French model, the chronic plus binge model, and the EtOH combined high-fat diet (HFD), as well as lipopolysaccharide (LPS) and other second hit models (Figure 1) [11]. The Lieber–DeCarli model usually only induces mild steatosis and elevated serum alanine aminotransferase (ALT) and aspartate transaminase (AST) levels, with very little or no inflammation of the liver [12]. The Tsukamoto–French model can lead to severe liver steatosis and fibrosis, liver cirrhosis with focal necrosis and immune cell infiltration, and significantly increased ALT and AST serum levels [13]. The chronic plus binge model simulates the drinking patterns of heavy drinkers. It can induce neutrophil aggregation and activation, which can then induce severe liver injury [14]. Common second hits include a high-fat diet, CCl_4_, and LPS, and when most of these “second hits” are combined with alcohol, they can induce the occurrence of alcoholic hepatitis and liver fibrosis in experimental animals [15,16,17,18]. However, the existing animal models of ALD fail to summarize all of the damage characteristics of human ALD. Therefore, it is necessary to study the correlation between various animal models and the progress of human ALD.

We have summarized various models according to the different stages of ALD. This includes the advantages, limitations, and translational relevance of the pure alcohol model, which is currently the most commonly used ALD animal model, as well as the research progress of the second hit model, which has shown great potential. In addition, we also discuss the pathogenesis of ALD and summarize the results of liver injury (including changes in intestinal microbes) and common in vitro models of ALD (Figure 2). This will aid in our understanding of the alcohol-associated liver disease model and its mechanisms to better treat alcohol-associated liver disease.

## 2. Pathogenesis of ALD

The pathophysiology of ALD is complex, and the direct effects of ethanol and its metabolites (e.g., acetaldehyde) on the liver and other organs, as well as immune cell activation and inflammation triggered by alcohol intake, are drivers of the ALD course [5,8]. 

### 2.1. Programmed Cell Death (PCD) in ALD

Programmed cell death (PCD) in ALD is thought to play a central role in the progression of liver injury. The pathways involved in PCD include the apoptosis, necroptosis, pyroptosis, autophagy, and ferroptosis pathways [19]. Apoptosis is a cystein-dependent cell death, and chronic alcohol intake induces oxidative stress, hypoxic stress, and endoplasmic reticulum stress in cells, leading to the activation of endogenous and exogenous apoptosis [20,21]. Cellular stress induces apoptosis in the liver by activating caspase-12/4, the interferon regulatory factor signalling pathway, and the C-Jun N-terminal kinase [22]. In addition, ethanol promotes inflammatory-factor-mediated liver cell death by activating proinflammatory responses in immune cells [23]. Ethanol exposure increases intestinal permeability and subsequent entry of LPS into the portal vein circulation, which stimulates resident hepatic macrophages, and infiltrates monocytes to produce inflammatory factors such as tumour necrosis factor -α(TNFα) through the activation of Toll-like receptor 4 (TLR4), thereby activating the exogenous pathway of apoptosis [22]. Necrosis is a novel form of PCD mediated by receptor-interacting protein kinase 1 (RIPK1), RIPK3, and mixed-lineage kinase domain-like pseudokinase (MLKL) [21]. Necrosis has been shown to play a key role in oxidative stress, steatosis, and inflammatory response in ALD [24,25,26,27]. Following ethanol ingestion, ROS production promotes RIPK1 activation, leading to RIPK3 recruitment and necroptosis, while RIPK1 and RIPK3 can mediate the reactivation of necrosis by ROS production, forming a positive feedback loop [28]. There is increasing evidence that RIPK3 and MLKL have been shown to be regulators of lipid metabolism, controlling hepatic steatosis to varying degrees [29,30]. With alcohol exposure, necrotic cells release danger signals called damage-associated molecular patterns (DAMPs) that trigger and promote inflammatory responses [21]. Pyroptosis is a newly discovered form of PCD mediated by gasdermin protein and is accompanied by inflammatory and immune responses [31]. Notably, chronic ethanol intake activates pyroptosis in both hepatocytes and hepatic immune cells, exacerbating hepatic inflammation through crosstalk between different cell types [20]. Both gut-derived sociated molecular patterns and metabolite-derived endogenous danger-associated molecular patterns (uric acid and ATP) are induced in ALD, causing the immune cells to release inflammasome-dependent cytokines to trigger pyroptosis [32]. Ethanol exposure can also overexpress thioredoxin-interacting protein (TXNIP) by stimulating liver cells, which, in turn, activates NLRP3 inflammasomes and caspase-1-mediated cell typical pyroptosis [33]. Autophagy, a mechanism of cell survival under conditions of nutritional stress, has also been implicated in the progression of ALD [34]. It is noteworthy that autophagy regulation is different between acute and chronic ethanol exposure [35]. Autophagy protects the liver from alcohol-induced acute ethanol attack and reduces lipid accumulation through phagocytosis [35]. However, chronic alcohol exposure inhibits mTOR activation, thereby inhibiting the initiation of autophagy [35]. In addition, chronic alcohol exposure increases lysosomal pH and inhibits the expression of the transcription factor TFEB, which is required for lysosomal fusion and autophagy, and these changes result in the inhibition of autophagy and lysosomal fusion in hepatocytes [36,37,38,39]. In addition, long-term alcohol consumption increases the serum ferritin concentration and transporter saturation, thereby increasing hepatic iron storage and the production of reactive oxygen species and causing local inflammation [19,40]. Firstly, oxidative stress mediated by ethanol exposure can inhibit hepcidin transcription by suppressing C/EBPα [41]. Secondly, ethanol exposure induces the expression of hypoxia-inducible factors and promotes elevated erythropoietin, which in turn inhibits hepcidin expression [42]. In addition, ethanol may also induce iron accumulation in the liver by acting on the transferrin receptor [43]. Notably, the DAMPs released from ferroptotic hepatocytes activate the NOD-like receptor family containing the NLRP3 inflammasome in KCs, and the release of proinflammatory cytokines and chemokines is concomitantly increased [44,45].

### 2.2. Adipose Tissue Damage

Adipose tissue is primarily considered a major storage organ, and alcohol intake can stimulate liver fat decomposition and promote liver fat regeneration, causing more lipids to enter the liver, resulting in steatosis, inflammation, and even adipocyte death [46,47,48,49]. Adipose tissue dysfunction such as adipocyte death and the inflammatory release of free fatty acids (FFA) may affect hepatic metabolism [50]. The process by which other tissues hydrolyse triglycerides to produce FFA during inflammation is called lipolysis [51]. A possible mechanism to increase lipolysis in adipose tissue after chronic ethanol ingestion is by affecting fibroblast growth factor 21 [52]. In addition to this, alcohol also alters the adipogenic pathway by upregulating sterol regulatory element-binding proteins (SREBPs) and inactivating peroxisome proliferator-activated receptor-alpha (PPARα) [53]. White adipose tissue (WAT) releases adipokines such as adiponectin leptin and other adiponectin, and therefore WAT has recently been regarded as a major secretory organ [52]. These secretions are negatively regulated in ALD and activate KCs and hepatic stellate cells (HSCs), leading to hepatitis and fibrosis [54,55,56,57]. In addition, the negative regulators of uncoupling protein 1 (Ucp1) and insulin signalling (PTEN and SOC3) can be upregulated after chronic alcohol consumption, causing WAT metabolism disorder and the activation of adipose triglyceride lipase and hormone-sensitive lipase [48]. Apart from this, chronic ethanol exposure affects the differentiation of liver adipose-tissue-derived stromal cells and the secretion of important regulators of lipid metabolism, such as TNF-α, interleukin-6 (IL-6), and monocyte chemotaxis protein-1 (McP-1) [47,50,58]. Finally, the proinflammatory state formed in WAT leads to lipodystrophy, which leads to fat deposition in the peripheral organs, thereby enhancing the pathological state of ALD [52].

### 2.3. Intestinal Dysbiosis

Recent evidence indicates that the gut is a key site for ALD [59]. Drinking can change the composition of the microbiota and its metabolites, causing an imbalance in the intestinal microflora and impairing intestinal integrity and barrier function [19,60,61]. Alcohol intake can modulate the glycosylation of mucins, thereby altering the protective mucus layer and potentially altering the bacterial species that adhere [62]. In addition, alcohol can reduce the expression of zonula occludens-1, increase the expression of Clock and recombinant period circadian protein 2 in Caco-2 cells, and upregulate cytochrome P450 2E1(CyP2E1) through reactive oxygen species (ROS), leading to excessive intestinal permeability [63,64]. Intestinal bacteria are significantly altered in patients with ALD, with alcoholics having significantly reduced numbers of *Enterococci*, *Bifidobacteria*, *Eubacterium g23*, *Oscillibacter,* and *Clostridiales* [65,66]. *Bacillus* and *Veillonella* were increased in the faeces of patients with severe alcoholic hepatitis relative to healthy subjects [66]. *Enterobacteriaceae* were 27 times more abundant in the faeces of patients with alcoholic cirrhosis than in healthy volunteers, and Enterobacteriaceae were the most common hepatic translocating bacteria in patients with cirrhosis [67]. Long-term alcohol intake can reduce the expression of recombinant regenerating islet-derived protein 3b, regenerating islet-derived protein 3g, and the secretion of c-type lectin, which is conducive to bacterial translocation and leads to the instability of the intestinal environment [68,69]. When the intestinal barrier is dysfunctional, high levels of TNF-α are produced by monocytes and macrophages within the intestinal lamina propria, a process that leads to the disruption of tight junctions between intestinal cells and increased intestinal permeability [70]. In the presence of increased intestinal permeability, the dysbiosis of the intestinal flora may impair hepatic homeostasis by increasing the levels of bacterial products or by the translocation of bacteria into the blood and lymph nodes or through the portal vein into the liver [69,71,72,73]. Bacterial translocation is a key pathological event in the transition from alcoholic steatosis to ASH and can also cause inflammation and hemodynamic disorders in patients with liver cirrhosis, leading to serious infections [19,74,75]. Studies have shown that the increased translocation of microorganisms such as *Streptococcus*, *Shuttleworthia*, *Rothia,* and *Nubsella* in the duodenal mucosa is associated with the early stages of progressive ALD [76]. In addition, the translocation of the components of intestinal bacteria (e.g., peptidoglycan, LPS, or flagellin) plays an important role in the progression of ALD, with changes in LPS being extensively studied [77,78]. LPS causes endotoxaemia through translocation and interacts with TLR4 in the liver, and the activation of the LPS-TLR4 signalling pathway promotes the release of proinflammatory factors (e.g., TNF-α and IL-6) and exacerbates alcohol-induced liver inflammation [70,78].

### 2.4. Changes in Immune Cells

The infiltration of inflammatory cells such as macrophages and neutrophils is a prominent feature of the early staging of ALD and ASH [79,80]. Hepatic macrophages from ALD patients have been documented to accumulate within the portal tract and function as regulatory signals in the immune microenvironment of the liver [19]. Kupffer cells (KCs) are inherent macrophages that are distributed in liver tissues [81]. The sensitization of portal vein LPS by hepatic macrophages is believed to be a key mechanism of steatosis, liver injury, inflammation, and fibrosis in ALD [82]. Mechanistic studies have shown that the triggering of the CD14/TLR4 receptor complex on KCs by LPS triggers the downstream IL-1 receptor-associated kinase (IRAK) to produce IL-1β, which subsequently recruits and activates liver iNKT cells (invariant natural killer T cells) and triggers the inhibitor of the nuclear factor-κ kinase (IKK) pathway, leading to the release of inflammatory cytokines as well as chemokines [83,84,85]. These inflammatory cytokines and chemokines enhance ALD inflammation and alcohol-induced liver injury. KCs activation often leads to oxidative stress, which leads to ALD organelle stress-sensitive liver cell damage, and a macrophage-induced increase in granulocyte colony-stimulating factor mediates neutrophil production/release [86,87]. Furthermore, macrophages activate T helper cells to release IL-17 by producing IL-23, which, in turn, mediates neutrophil production/release through an increase in granulocyte colony-stimulating factor [88]. Neutrophils are the most abundant innate immune cells in the human body, and their functions in ALD are complex and diverse [19]. Neutrophils can directly contribute to the development of inflammation and hepatocyte injury in ALD, while the cytokines released by neutrophils are important mediators regulating inflammation and tissue repair [89,90,91]. Additionally, neutrophils regulate the gut microbiota and bacterial infection in ALD by killing and phagocytosing pathogenic microorganisms [92]. T cells and NKT cells are also involved in the pathogenesis of ALD. T cells not only promote disease progression by releasing inflammatory mediators such as TNF-α, IL-1, and IL-17 but also directly damage liver cells through cytotoxic CD8+T lymphocytes and play a beneficial role in ALD by reducing inflammation and promoting liver regeneration [10,93,94,95]. NKT cells in the liver can directly recognize lipid antigens through surface receptors and TCRs or indirectly activate APCs (such as KCs, hepatocytes, and myeloid DCs) to secrete TLR ligands and cytokines (such as IL-12, IL-4, and IFN-γ) [94,96,97]. In addition, type I NKT cells induce inflammation and neutrophil recruitment, leading to liver tissue damage, while type II NKT cells have a protective effect against ALD damage [98]. In future studies, targeted interventions and treatment strategies should be developed based on the complexity and diversity of immune cells function.

### 2.5. Oxidative Stress

In addition to the aforementioned mediators, several other factors have also been shown to play important roles in the pathogenesis of alcohol-associated liver disease. Notably, oxidative stress in the liver has been recognised as a hallmark feature of ALD, with highly reactive free radicals produced by ethanol and its metabolites promoting liver tissue damage [99]. After being absorbed through the gastrointestinal tract, only 2–10% of the total intake of alcohol is directly excreted through sweat, etc., in the form of ethanol [100]. Most alcohol is metabolized in the liver, and the major oxidative pathways involve ethanol dehydrogenase (ADH) and aldehyde dehydrogenase (ALDH) and cytochrome P450 in the microsomal ethanol oxidation system [5,100,101]. First, ethanol is oxidized to acetaldehyde by ADH in hepatocytes, a reaction that requires a reduction in nicotinamide adenine dinucleotide (NAD+) to NADH as an intermediate carrier [102]. After oxidation, most acetaldehyde is catalysed to acetate by NAD+/NADH via ALDH in mitochondria [103]. With the production in acetaldehyde and in the NAD+: NADH REDOX ratio, the glutathione (GSH) transport through the inner mitochondrial membrane is impaired, thus reducing the antioxidant reserve of cells [104]. Cytochrome P450, including CYP2E1, generates reactive oxygen species (ROS) through alcohol metabolism, such as hydrogen peroxide (H_2_O_2_), superoxide anion (O^2−^), and hydroxyethyl (·OH), and can react with acetaldehyde, leading to the formation of acetaldehyde protein adducts [105,106,107]. All of these changes increase the risk of tissue damage [107]. Acetaldehyde, on the other hand, is also responsible for the production of reactive oxygen species, which cause oxidative stress, endoplasmic reticulum stress (ER stress), and steatosis [108]. Hepatic oxidative stress also stimulates the development of hepatic inflammatory responses, creating a pathological cycle that promotes the progression of ALD [99]. Additionally, alcohol intake induces ER stress followed by the production of proinflammatory mediators and DAMPs that activate the nuclear translocation of cyclic AMP response element-binding protein H (CREBH) and nCREBH [10]. Another consequence of ER stress is the activation of lipogenic pathways in hepatocytes, particularly through SREBPs, followed by the upregulation of the predominant form of FSP27-β [109,110]. The interaction of FSP27 with lipid droplet membrane proteins promotes lipid droplet formation and steatosis. Furthermore, ethanol intake promotes the translocation of FSP27 to mitochondria and leads to mitochondrial damage and hepatocyte death [10]. All of these factors lead to hepatic neutrophil infiltration and liver inflammation.

## 3. Current In Vivo Models of ALD

Animal models can provide profound experimental strategies that are difficult to complete in clinical studies and can help to reveal the pathogenesis of human diseases. Rodent models have become the preferred experimental models for the preclinical studies of human diseases in many fields, including in ALD, due to their relatively low cost, shorter gestation time, shorter fecundity, and, perhaps most importantly, ease of handling of gene manipulation.

### 3.1. Lieber–DeCarli Model

The Lieber–DeCarli model is one of the most frequently used rodent experimental models in the study of early ALD (Table 1). The control liquid diet consists of a mixture of protein (15% of total calories), fats (36% of total calories), carbohydrates (49% of total calories), vitamins, and salts, with alcohol replacing 35.5 percent of the total calories allocated for carbohydrates in the ethanol-containing formula diet (LDE diet) [111,112]. The Lieber–DeCarli model consists of an initial domestication phase of approximately 7 days, with ethanol gradually increasing from 0 to the final concentration (5% ethanol *w*/*v*). After the acclimatization feeding is completed, feeding with the LDE diet is continued for 4–12 weeks [111]. When animals were freely fed the Lieber–DeCarli diet (5% ethanol *w*/*v*) for a long period of time, the blood alcohol concentration (BAC) of rodents reached 100~150 mg/dL and the plasma ALT and AST levels of the animals were significantly increased, which was accompanied by a six-fold increase in liver triglyceride levels, causing mild liver damage [13,113]. In this model, rodents can overcome their aversion to alcohol when they are given only a liquid diet containing ethanol without any edible food or drink. At this point, the rodents consume approximately 15 g/kg of ethanol per day [111,114]. The advantages of this model are that it is economical, is simple to operate, does not require special surgical skills or expensive equipment, and has a very low mortality rate [112]. As a mild long-term feeding mode, it is often used in the early stages of ALD research [13].

It is worth noting that the drinking pattern of this model is different from human drinking habits because the rodents are forced to drink alcohol when they are hungry and thirsty and they switch from eating solid food and water regularly to a liquid diet, thus inducing some changes in physiological phenomena [13,115]. In addition, this model can only induce mild steatosis in rodents but rarely causes liver inflammation and fibrosis and fails to induce advanced ALD (cirrhosis and HCC), which has obvious limitations in the reproduction of ALD progression [12,115,116].

### 3.2. Tsukamoto–French Model (TF Model)

The Tsukamoto–French model overcomes the rodents’ natural aversion to alcohol and allows for the better control of alcohol consumption in animals, resulting in more severe liver damage [13]. In this model, an implanted gastric tube and fluid pump are used to infuse the animals with a liquid diet containing ethanol daily, with the ethanol accounting for 49% of total calories [117,119]. After 1 month of infusion (22–35 g/kg/d), the animals developed severe hepatic steatosis and focal necrosis, with a mean BAC of about 300 mg/dL, accompanied by a significant increase in their ALT and AST levels [123]. Notably, the TF model also allows for the easy manipulation of nutrient content to create the desired liver injury model. Dietary ethanol (32–47% of total calories) combined with polyunsaturated fats (25% of total calories) led to the further development of steatohepatitis in rodents, with fibrosis beginning to be observed within 30 days and liver fibrosis in all animals after four months [120]. In 2015, the Tsukamoto–French team developed a new hybrid model based on the original alcohol gavage model. After the animals had been fed a Western diet (high in cholesterol and saturated fat) for 2 weeks, a gavage catheter was implanted for ethanol infusion. During the 8-week model period, the ethanol intake was gradually increased to 27 g/kg per day, and from the second week onwards, alcohol binge (4–5 g/kg) was carried out once a week [7]. Among them, the repeated administration of the ethanol to the animals triggered a transition from chronic ASH to acute AH. This model shows the clinical features of alcoholic hepatitis, such as splenomegaly, hypoalbuminemia, and hyperbilirubinemia, for the first time [7]. 

In conclusion, the TF model induces a liver injury process similar to human ALD in rodents, including progressive steatosis, fibrosis, and liver cirrhosis characterized by focal necrosis and immune cell infiltration [13,120]. While the TF model did cause more severe liver damage compared to other models, there are several potential drawbacks. The surgical insertion of gastric tubes into the animals and the subsequent months of care require a high level of technical and maintenance [13]. Its technical difficulties and the large amount of equipment required make the TF model expensive and unable to be implemented in all laboratories [121,124].

### 3.3. The Chronic Plus Binge Model

The original chronic plus binge model (known as “Gao-Binge” or the NIAAA model) was developed by Gao Bin’s team [141]. The pattern includes single or multiple binge drinking, similar to the drinking pattern of many alcoholic hepatitis patients who have a background of long-term (chronic) drinking and a recent history of excessive drinking [125]. In 2010, a short-term plus binge ethanol feeding model was applied to mice for the first time [141]. In this model, the mice were fed the Lieber–DeCarli control diet for 5 days followed by the LDE diet (containing 5% ethanol) for 10 days, while the BAC was approximately 180 mg/dL. On the 16th day, the mice were administered a single dose of ethanol (5 g/kg, 20% *v*/*v*) by gavage [115,141]. The highest BAC level was 400 mg/dL after a single binge. The mice were euthanized after 9 h of alcohol binge drinking, and the peak levels of ALT and AST were approximately 250 IU/L and 420 IU/L, respectively, accompanied by obvious liver injury [11,141]. In 2013, the model was incrementally increased with ethanol (1–5% of liquid diet) over 5 days of adaptive feeding, and the concentration of the ethanol solution was adjusted to exceed 31.5% (*v*/*v*) by in-gastric administration [115]. In 2015, the team improved the ethanol chronic plus binge feeding model. During chronic feeding, the mice were fed with the LDE liquid diet from 10 days to 12 weeks combined with a single insufflation (5 or 6 g/kg) or LDE diet for 8 weeks plus multiple binges of ethanol (5 or 6 g/kg, twice a week for 8 weeks) [110]. Compared to the previous 10-day LDE plus one alcohol binge feeding model, increasing the duration of chronic feeding and the frequency of binge drinking periods caused more severe ASH and mimicked some aspects of early steatohepatitis in AH patients, including serum ALT and elevated AST levels, steatosis, neutrophil infiltration, and mild barbed wire fibrosis [110]. In recent years, the NIAAA model was improved. After chronically feeding mice with the LDE diet for 4–7 weeks, combined with a single binge (5 g/kg) or multiple binges (5 g/kg, 31.5% *v*/*v*, 3 doses, 12 h apart), the improved model increased the animals’ BAC levels, ALT and AST levels, inflammation, and neutrophil infiltration [112]. In addition, it was found that when mice of different ages were used for the NIAAA model, the aged mice (>16 months) were more susceptible to liver damage, inflammation, and even fibrosis caused by alcohol overdose, which may be related to the fact that aging can aggravate the process of ALD by downregulating the SIRT1 in hepatic stellate cells, hepatocytes, and neutrophils [128,142]. In a rat model of chronic plus binge drinking, ethanol concentrations were increased from 1.25% to 5% during acclimation feeding, followed by 4 weeks of the LDE diet, in combination with repeated binge drinking (32% *v*/*v*, 3 gavages, 12 h apart) [141]. This model induces hepatic steatosis in rats, significantly increases BAC levels, and increases hepatic oxidative stress and proinflammatory cytokine production, leading to hepatic steatosis and inflammation [11,125,142]. This model significantly elevates BAC levels and increases liver oxidative stress and the production of proinflammatory cytokines, leading to liver steatosis and inflammation [11,126].

In the chronic plus binge model, alcohol binge can accelerate the metabolism of alcohol and the development of its metabolite acetaldehyde in the liver within 2–3 h [127]. Earlier studies have found that when compared to long-term or binge drinking alone, the chronic plus binge model can significantly upregulate the expression of IL-1β and TNF-α in the liver and induce the accumulation of liver neutrophils, but it does not induce macrophage infiltration [14]. It is worth noting that recent studies have shown that this model can also increase the number of iNKT cells in the liver and induce their activation [83,143]. 

The chronic plus binge drinking model reproduces the drinking behaviour and acute/chronic liver injury of ALD patients and has been widely used to study the pathogenesis of ASH and mild AH [10,115]. In the chronic plus binge drinking model, the short-term model is simple to manipulate and inexpensive and reproduces ASH well but does not induce fibrosis. In the long-term model of 8–12 weeks, significant steatosis, inflammation, and mild fibrosis can be induced, but the economic and care costs of the model increase due to the relative effort involved in long-term feeding.

### 3.4. Second Hit Models

If the pathology of human ALD is to be reproduced more accurately, it is necessary to provide a second hit in an animal model. In current research, well-known second hits include nutritional modification, an agonistic/xenobiotic second hit, viral infection, and a genetic second hit [15].

#### 3.4.1. Nutritional Second Hit

Nutritional modification is one of the most common environmental secondary attack methods. Both a high-fat diet and excessive alcohol consumption initially contribute to the development of hepatic steatosis, and combining a high-fat diet (HFD) with alcohol consumption is one of the most basic nutritional regulation approaches [15,130]. In the model developed by Chang, mice were fed an HFD for 3 days or 3 months followed by a single dose of ethanol gavage on the last day (5 g/kg, 3d-HFD+ 31.25% ethanol or 3m-HFD+ 53% ethanol) [129]. Studies have shown that short-term and long-term HFD plus acute ethanol binge can lead to significant liver neutrophil infiltration, reduced liver macrophages, and increased ALT and AST levels in mice. The trend for the long-term models is more obvious [11,129]. In addition to this, this study suggests that obese alcoholics may be more likely to advance from steatosis to advanced ASH [18]. The model successfully simulates acute steatohepatitis in obese alcoholics but requires a longer modelling period.

In addition, the liver plays an important role in maintaining iron homeostasis in the body [144]. In the iron plus alcohol intake model, the carbonyl iron method mainly implements iron loading in hepatocytes, in which iron catalyses and promotes liver oxidative stress and damage [15,118]. Accordingly, hepatic iron overload is prone to the development of ALD [145]. In the rodent ALD model, the second hit of iron significantly aggravated alcoholic liver damage and promoted the formation of alcoholic liver fibrosis [15]. Giving rats a long-term implanted gastric catheter and adding iron carbonyl (0.12% *w*/*v* in the first week and 0.25% *w*/*v* after the second week) to a high-fat diet (25% of total calories) combined with an ethanol (49% of total calories) diet, after 16 weeks of feeding, the serum levels of ALT and AST in rats rose to 2–3 times those of rats fed a normal diet, resulting in moderate to severe fatty liver as well as focal lobular central necrosis and inflammation, with some animals developing liver fibrosis and even cirrhosis [118]. As this model uses a gastric catheter to ingest the diet, the cost of care and the required equipment is high [13]. However, it has the advantage of overcoming the animals’ natural aversion to ethanol and inducing ASH, AH, liver fibrosis, and even cirrhosis within 16 weeks.

#### 3.4.2. Agonistic/Xenobiotic Second Hit

Common exogenous stimuli for the second hit include LPS, CCl_4_, acetaldehyde compounds, acetaminophen (APAP), and N-Nitrosodiethylamine (DEN). The original LPS plus ethanol model was developed by Bhagwandeen et al. Rats were fed an LDE diet for 10 weeks and then injected intravenously with LPS (10 mg/kg). Chronic ethanol combined with low-dose LPS can cause hepatocyte necrosis and neutrophil infiltration in rats with severe steatosis. The final BAC level of rats was approximately 90 mg/dL [16]. In recent years, a variety of alcohol binge drinking models combined with LPS have been established. When mice were administered alcohol (6 g/kg) by gavage for three consecutive days and the intraperitoneal injection of LPS (10 mg/kg) was performed 24 h after the last alcohol gavage, the degree of liver necrosis and hepatic neutrophil infiltration as well as the levels of ALT and AST were significantly increased (~400 U/L) [131]. In another model, ALT levels were also significantly increased after the mice were administered an LDE diet for 4 weeks, followed by one intragastric administration of ethanol (5% *v*/*v*) and an intraperitoneal injection of LPS (2 mg/kg). Mouse liver sections also showed lipid droplet accumulation and enhanced liver damage, with distinct areas of necrosis and inflammatory cell infiltration [132]. In conclusion, the LPS plus ethanol model can induce significant lipid accumulation, inflammation, and liver damage in rodents in a short period of time [146]. Moreover, this model has the advantage of being inexpensive and easy to implement in most laboratories.

The ethanol plus CCl_4_ model for alcoholic liver fibrosis is considered to be a classic model. The mice are injected with CCl_4_ (0.5 μL/kg, once every 3 days) in the intraperitoneal cavity within eight weeks, and this is combined with an LDE diet [17,133]. This model shows a correlation pattern similar to that of human alcoholic cirrhosis while showing the accompanying process [133,134]. In a recently developed ethanol plus CCl_4_ model, liver damage was caused in mice through the inhalation of CCl_4_ (once a week for the first 4 weeks and twice a week for the following 3 weeks), and ethanol was added to the drinking water (4% during the first week, 8% during the second week, and 16% until the final week). 

The inhalation of CCl_4_ can cause significant fibrosis within 4 weeks and a strong proinflammatory reaction within 7 weeks [135]. The model-induced fibrosis, inflammation, and steatosis in the mice was similar to human alcoholic liver fibrosis [136]. In addition, Wonhyo Seo et al. successfully induced HCC using an alcohol diet plus CCl_4_. Severe liver fibrosis was first induced in mice by the intraperitoneal injection of CCl_4_ (0.2 mL/kg olive oil, twice weekly) for 18 weeks, followed by a combination of an LDE diet (4% *v*/*v*) and CCl_4_ injection for a further 10 weeks. After 28 weeks of alcohol plus CCl_4_ administration, liver histopathology in mice showed steatosis, inflammation, fibrosis, the ballooning of hepatocytes, and tumour nodules [137]. The ethanol plus CCl_4_ model successfully reproduces most of the disease course and liver pathology of ALD, and although it is time-consuming, it is simple to operate and inexpensive to care for and is a common model for alcoholic liver fibrosis.

In addition, in the ethanol plus APAP model, mice were gavaged with ethanol (4 g/kg of ethanol every 12 h for 2.5 d), and 12 h after the last binge, they were injected with APAP (150 and 300 mg/kg) by oral gavage [138]. In another study, mice were fed an LDE diet containing 5% EtOH (5% *v*/*v*) for 15 days (including 5 days of acclimatization feeding) and on the last day were given the corresponding EtOH gavage (5 g/kg, 31.5% *v*/*v*) and an intraperitoneal injection of APAP (200 mg/kg). The study showed that this model increased inflammatory secretion, lipid accumulation, and oxidative stress in mice, accompanied by a significant increase in ALT and AST levels [147]. This ethanol plus APAP model can reproduce the characteristics of early ALD in a short period of time and is commonly used in studies of drug hepatotoxicity.

DEN is an alkylating agent of DNA bases that induces gene expression patterns in mouse tumours that resemble the poor prognosis subclass of human liver cancer [148,149]. Therefore, DEN combined with an LDE diet is often used to feed mice to reproduce the HCC model. The specific protocol is to inject 75 mg/kg of DEN intraperitoneally every week for the first three weeks and then to adjust the DEN dose to 100 mg/kg for the next three weeks followed by 7 weeks of the LDE diet. During this process, liver damage continued to increase in mice and eventually showed an increased recruitment of precancerous liver macrophages with a mixed M1/M2 phenotype [150]. Another relevant model was an intraperitoneal injection of DEN (10 mg/kg) in 2-week-old mice. Mice were then administered an LDE diet (4.8% alcohol) at 3 months of age for 3–7 weeks. In this model, ethanol plus DEN induced visible superficial tumours in mice, and the serum alpha-fetoprotein levels increased to three times those of normal mice [140]. The ethanol plus DEN model is simple to operate, inexpensive, and histologically and genetically reflective of alcohol-induced HCC, making it one of the commonly used models for HCC.

#### 3.4.3. Viral and Genetic Second Hit

The genetic second hit includes two aspects: enhancing the function of suspected pathogenic genes to promote the damage mechanism and the reduction in or loss of the function of protective genes that make the liver sensitive to the harmful effects of alcohol [15]. At present, hepatitis B virus (HBV) and hepatitis C virus (HCV) are still the most important risk factors for HCC worldwide [151,152]. According to the research, HCV-positive patients who drink alcohol will develop from liver fibrosis and cirrhosis to HCC faster than those who do not drink alcohol [153,154]. These two factors synergistically affect the immune response, cytotoxicity, and oxidative stress to accelerate a series of events leading to liver cirrhosis and hepatocellular carcinoma [155,156]. Ethanol was added to the drinking water of 2-month-old HCV core transgenic mice at an initial concentration of 5%, which was increased by 5% to 20% every two weeks until the mice were 10 months old. The 10-month-old mice were administered water and a 25% ethanol solution (ethanol intake was 2.5 g/kg) through gastric intubation for 24 h. The results showed that alcohol consumption increases hepatic lipid peroxidation and synergizes with the HCV core protein to increase hepatic TGF-β and TNF-α gene expression, indicating that the model simulates the accelerated development of fibrosis observed in HCV-infected alcoholics [154].

Correspondingly, HBV infection is also a risk factor for hepatocellular carcinoma and liver-related death in patients with ALD [157]. HBV-Tg mice containing the intact HBV genome were fed a Lieber–DeCarli liquid diet containing ethanol (5% *v*/*v*) for 4 weeks. The ethanol content of the liquid diet was then adjusted to 7% *v*/*v* for the next 4 weeks. Eight weeks of feeding resulted in increased lipid droplets in the liver tissue of the mice, accompanied by increased triglyceride and cholesterol levels. It was shown that HBV and alcohol diet synergistically induced abnormal hepatic lipid metabolism in mice, leading to alcoholic fatty liver [158].

Although second-hit models are well suited to induce early- and late-stage ALD in animals, there are differences in the pathogenesis of liver injury caused by the combination of ethanol and a second liver-damaging substance, compared to liver injury caused by ethanol alone [9]. It is preferable for the second hit to be “natural”, as a model that knocks out a gene or includes excessive drug treatment will not fully reproduce the characteristics of human ALD. Future research requires more second-hit models that are similar to the human natural living environment to help understand the pathogenesis of ALD and discover new therapeutic targets.

### 3.5. Other Animal Models of ALD

Nonhuman primates (monkeys and apes) have long been regarded as important laboratory animals for the study of biomedical disease processes [159]. Because they are similar to humans in genetics, anatomy, behaviour, and physiology, they can be used to bridge the translational gap between rodent and human research [160,161]. In an initial study, baboons were given ethanol (36% of total calories) as part of their drinking water and maintained on an adequate diet by being fed special high-protein biscuits, but even after 3 years only steatosis could be observed and no alcoholic hepatitis or cirrhosis was detected [162]. The researchers therefore adjusted the alcohol intake in the liquid diet to 50% of total calories, combined with protein (18% of total calories), fat (21% of total calories), carbohydrate (11% of total calories), and a moderate amount of vitamin supplements [162]. Baboons consumed an average daily diet of 81 ± 3 mL/kg and developed alcoholic hepatitis after 9–12 months on an oral liquid diet, with marked hepatic steatosis, hepatitis, and extensive fibrosis accompanied by TC levels up to 205 mg/kg at 20–21 months and complete cirrhosis after 24 months on an alcoholic diet and BAC up to 260–367 mg/dL [163]. Additionally, the study showed that the baboons in this model did develop severe liver lipid deposition under self-feeding conditions. This suggests that differences in alcohol-induced hepatic lipid deposition thresholds appear to be alcohol intake in the form of a liquid diet (lower incidence) or self-administration (higher incidence) [161,164]. In a recent study, a model of ethanol self-administration in nonhuman primates was used [165,166]. The animals were first induced to self-administer EtOH by increasing the dose of EtOH in the drinking water by 0.5 g/kg every 30 days for three months, from 0 g/kg/d to 1.5 g/kg/d. In this way, all rhesus monkeys had a BAC of over 50 mg/dL. After a 3-month ethanol induction period, the animals were given “open access“ to ethanol (4% *w*/*v*) and water for 22 h per day for 24 months [165]. In this model, animals developed ALD at 18 months into the diet, exhibiting steatosis and liver inflammation accompanied by elevated liver enzymes (including ALT, AST, and alkaline phosphatase) and significant oxidative stress [165,167,168]. Nonhuman primate models can show pathophysiological processes similar to those in humans and can show ASH, AH, and liver fibrosis and even complete cirrhosis in long-term models [163,167]. However, these animals are somewhat aggressive after alcohol consumption and the models are too time-consuming and expensive to use in many laboratories, both economically and in terms of care [161].

In addition, the zebrafish has emerged as a powerful vertebrate model for studying liver-related diseases. Although the structure of the zebrafish liver is different from that of mammals, it is functionally similar to 70% of human disease genes and exhibits very similar basic physiological processes, genetic mutations, and pathogenic responses to environmental damage consistent with humans [169]. Zhou et al. used transgenic zebrafish larvae (Tg (lfabp10α: eGFP)) at 96–98 h post-fertilisation, exposed to a 2% ethanol solution and incubated for 32 h at 28 °C in an incubator. Zebrafish larvae showed hepatomegaly lordosis and emaciation, accompanied by severe lipid accumulation and oxidative stress in the liver tissue [170]. In addition, the transcription levels of genes related to fatty acid metabolism and balance in the liver tissue of zebrafish larvae were altered and alcohol metabolism was accelerated, resulting in liver damage and reduced toxic metabolism [169]. In another study, 8-month-old adult male zebrafish were treated with water containing 0.2% ethanol (*v*/*v*), replacing the ethanol-containing water in the incubator every morning. After 4 weeks, histological analysis revealed ballooned hepatocytes and mild fibrosis in low-dose ethanol-treated zebrafish, accompanied by high ALT levels (200 IU/L) and high triglyceride levels (300 mg/dL) and elevated BAC levels to 0.175% [169,171]. Compared to rodent and nonhuman primate models, zebrafish have a different liver anatomy and framework, and their less conservative physiology and morphogenesis are major drawbacks to their use in biomedical research [169,172]. However, because of its high reproductive rate, affordability, and ease of maintenance, and the convenience it offers in terms of genome editing, with fewer ethical constraints, zebrafish have a great advantage in the study of ALD models [173,174].

## 4. In Vitro Model

Most of what we currently know about liver disease in humans comes from animal models. However, animal models are not able to fully reproduce human ALD process, highlighting an area where appropriate in vitro models can complement ALD models to further understand key developmental and pathological mechanisms.

### 4.1. Two-Dimensional (2D) Monolayer Cell Cultures Model

In the past 50 years, two-dimensional (2D) monolayer cell cultures have commonly been used in in vitro models to study cell biology [175]. The composition of the liver can be divided into two main types of cells: epithelial-derived parenchymal cells, namely hepatocytes and bile duct cells, and nonparenchymal cells, which are mainly composed of KCs, hepatic endothelial cells, and hepatic stellate cells [176,177]. Ethanol, LPS, and palmitic acid (PA) are commonly used stimuli when culturing these cells to construct an in vitro ALD model (Table 2). When EtOH was used as a stimulus, lipid accumulation was induced in AML-12 cells and HepG2 cells after incubation with 100 mM ethanol for 24 h, resulting in an increase in cellular TG levels, fatty acid synthase (FASN), and sterol regulatory element-binding protein-lc (SREBP-1c) mRNA expression [178,179,180]. In addition, ethanol can also induce severe oxidative stress by inducing CYP2E1 overexpression in HepG2 cells, which is manifested as a decrease in glutathione (GSH), an increase in ROS in whole-cell lysates, and superoxide in the mitochondria [179,181]. In previous studies, hepatocytes and liver macrophages were exposed to 100 mM ethanol and 50 mM ethanol for 48 h and then treated with 500 ng/mL LPS for 6 h. This model can cause cellular apoptosis, mitochondrial damage, and the autophagy response [146,182,183]. EtOH (1 or 5 μg/mL) and PA (400 μmol·L^−1^) are added to the culture medium to treat the cells and are left for 24 h in vitro to imitate the HFD plus EtOH model. In this model, EtOH promotes PA-induced hepatic lipid accumulation and inflammation in rat primary hepatocytes by activating the ER stress response [74]. 

With the increasing demand for in vitro models, cell culture methods are developing towards more realistic imitation. A 2D sandwich structure is a major innovation of 2D culture technology. Embedding primary human hepatocytes (PHH) into two layers of extracellular matrix (ECM) proteins provides a stable scaffold that allows hepatocytes to improve cell-to-cell contact by interacting with two layers of ECM proteins [184]. However, the scarcity and logistically difficult nature of PHH has prompted researchers to explore alternative cell sources. In recent years, continuous advances in the stem cell field have offered hope for the generation of hiPSC-derived hepatocytes as a potential alternative source of PHH, but great efforts are required to develop standardized protocols [185,186]. hiPSCs have great potentials as sources of mature hepatocytes. Woo et al. showed that lithium-treated embryonic stem cells (ESCs) differentiated into cells with hepatocyte-like morphology and expressed albumin and keratin in response to a combination of hepatocyte growth factor, tumour suppressor M, and dexamethasones [187]. Furthermore, Takebe et al. reported that hiPSC-derived hepatic endoderm cells were cultured with human umbilical vein endothelial cells and human mesenchymal stem cells to form a three-dimensional spherical tissue mass (induced pluripotent stem-cell-derived liver buds) that expressed the early liver markers gene [188]. 

The 2D monolayer cell culture is inexpensive and easy to maintain and is the most widely used method for the in vitro modelling of the liver as an alternative to animal models for drug safety and screening for early drug candidates [189]. Nevertheless, current 2D cell models are limited to the short-term analysis of hepatotoxicity. This is because even when cultured in an improved 2D sandwich, the 3D environment of the ALD liver cannot be accurately reproduced due to the short lifespan and dedifferentiation process of the cells [190].

### 4.2. Three-Dimensional Cell Culture and Liver-on-Chip Model

The complex structure of the liver cannot be completely simulated by monolayer cell culture systems. Therefore, combining different cell types in a functional three-dimensional (3D) structure can overcome the shortcomings observed when liver cells are cultured in 2D structures [197]. This is crucial for coordinating the liver’s specific response to injury and toxic compounds. In recent years, organoid technology has emerged as an in vitro 3D system to reproduce tissues in petri dishes, and liver organoids have been successfully established from primary liver tissues [198,199]. Recently, Wang et al. developed an in vitro organoid model system that can reproduce the typical characteristics of ALD pathophysiology by integrating human foetal liver mesenchymal cells (hFLMCs) into expandable hepatic organoids (hEHOs). The team used hFLMCs/hEHOs to prepare an in vitro ALD model, and ethanol (100 mM) was added to the culture medium and cultured for 7 days. Compared to the untreated control group, the ethanol-treated organoid group showed several characteristics of alcohol-induced liver injury, such as obvious increases in CYP2E1 and CYP3A4 activity, extracellular matrix deposition, apoptosis, and oxidative stress [191,192]. The 3D cell culture system is simple to operate, provides a microenvironment that is more similar to the living conditions in vivo for cells, and displays cell activities such as differentiation and intercellular reactions. Therefore, it can realize the real cell biology and function, accurately establish the target tissue model, and effectively predict the course of disease and drug response [190].

In a recent study, a new in vitro ALD model was successfully constructed on a chip. The chip contained cocultured spheres of primary rat hepatocytes and hematopoietic stem cells. The spheroids were exposed to a medium containing ethanol (60 mM) for 48 h to demonstrate the damage caused by alcohol [193]. In recent years, a team has developed a detachable on-chip liver device. The composition of this novel model included HepG2, LX-2, EAhy926, and U937 cells. It is worth noting that this model can improve the activity of HepG2 cells and reproduce the process of alcohol damage to hepatic nonparenchymal cell lines, which is conducive to understanding intercellular communication among various types of hepatic cells during the progression of ALD [194]. In order to simulate human ALD, Jang et al. used a liver chip of primary human cells. The composition of this liver chip included primary hepatocytes, primary hepatic sinusoidal endothelial cells (LSECs), and Kupffer cells. The ethanol concentrations of 0.08% to 0.16% were selected to simulate the BAC of human patients after drinking alcohol. Multimodal phenotype and function analysis was performed on the chip [195]. Based on this study, Janna C. Nawroth et al. found an increased release of hepatic lipid droplets, mitochondrial reactive oxygen species, and proinflammatory factors after perfusing the chip with 0.08% ethanol + LPS for 48 h [196]. Liver-on-chip models can be used to reconstruct 3D structures on a microscale to achieve physiological fluid flow and hepatocellular crosstalk, which can reproduce key features of the hepatocellular environment and achieve basic liver functions. Therefore, it has been widely used to study drug-induced hepatotoxicity and liver function and is currently an effective preclinical tool for tissue engineering and drug screening applications.

## 5. Discussion

In previous decades, the development of ALD animal models allowed for significant progress to be made in ALD research [13]. Different ALD models can be used to study the pathogenesis of ALD at different stages [135,140,200,201]. However, by comparing different ALD models, we found that in models only using an LDE diet, the main changes in animals were limited to early ALD (hepatic steatosis and inflammation), while those of more severe ALD (such as hepatitis and fibrosis) were rarely able to be observed [112]. Alcohol, as a susceptibility factor for many diseases, can affect the progression of the disease, and in many cases, this effect is only clinically relevant after secondary injury [13,202]. When the liver is damaged, alcohol drinkers are more likely to develop fibrosis and cirrhosis [203,204]. Correspondingly, when rodents receive a second hit other than alcohol, liver fibrosis and cirrhosis will also appear in the liver [15,129,132,133,138,139]. This is also why patients with end-stage ALD usually have end-stage viral hepatitis, diabetes, and other comorbidities [15]. However, because of the nonalcoholic effect in the second hit model, the results of the modelling may be different from the mechanism and pathological changes caused by alcohol alone [9,205]. 

Furthermore, although most alcohol-associated liver disease can be achieved in animals by altering stimulating factors and experimental animals, it is currently difficult to establish animal models of alcohol-triggered HCC that can reproduce pathophysiological processes similar to those of human HCC [206,207]. As a precarcinogen, alcohol-induced liver inflammation and the related oxidative stress can cause DNA damage in liver cells [208]. Long-term heavy drinking will support the occurrence and development of tumours through the production of carcinogenic aldehydes and reactive oxygen species [209]. Chronic drinking is an important contributing factor to HCC and coexists with chronic hepatitis virus infection as the main risk factor of HCC, and furthermore, they can also each promote the development of the other [152,210]. Thus, more attention should be paid to comorbidities and the development of animal models with synergistic pathological results caused by these comorbidities to better study the new mechanisms and therapeutic targets of ALD [15,206]. In addition, it is also possible to search for mouse models combined with human HCC genomic data to provide information about the different aspects of the onset and prognosis of HCC [137,211]. 

Another point worth noting is that the development of AFL into inflammation is a prerequisite for early ALD to develop into intermediate and late ALD (fibrosis, cirrhosis, and HCC) [212]. After it develops into fibrosis, reducing the alcohol supply or other treatments will not reverse the course of the disease [139]. However, for all stages of ALD, achieving abstinence is the best treatment strategy [213]. The use of drugs (e.g., disulfiram, naltrexone, and acamprosate) to reduce alcohol intake and treat alcohol dependence is an important strategy in the treatment of ALD [214]. In addition, ALD is associated with hepatocellular damage and liver regeneration disorders [212]. The application of many hepatoprotectants against oxidative stress and ROS production is also emerging to protect hepatocytes from damage and to promote liver regeneration [215]. Inflammation is also a key factor in promoting AH liver injury [216]. Treatment guidelines for EASL recommend the use of corticosteroids to reduce liver inflammation in patients with severe AH [217]. Currently, other anti-inflammatory drugs for the treatment of AH are also in clinical trials, including IL-1 inhibitors (such as anakinra), apoptosis signal-regulated kinase 1 inhibitors, probiotics, etc. [213,218,219]. In addition to this, epigenetic therapies are rapidly emerging as promising approaches for the treatment of ALD, including two histone-modifying enzyme inhibitors (HDAC inhibitors and DNMT inhibitors) and ncRNA-based therapies [220]. 

Furthermore, the species, gender, age, and environment of the animals may have an impact on the construction of the model [10,142,221]. More obvious damage can be obtained using the same conditions via choosing animals that are more sensitive to alcohol (such as elderly and female rats) to build a model [221,222]. Interestingly, some changes, such as increased room temperature, can significantly reduce mortality as well as the extent of liver damage in the rodents being greatly reduced [10,112]. Therefore, further rigorous research is needed to clarify the differences in the susceptibility of rodents to ALD in different environments [10]. At present, rodent ALD models are still ideal and effective tools for improving our understanding of ALD [205]. Although there are differences in the degree and stage of ALD between rodents and humans, with continuous improvements and development in ALD animal models, future models could gradually simulate each stage and the pathological characteristics of human ALD [9]. A better understanding of pathological damage by constructing ALD models at different stages could also better help to prevent and treat disease.

## Figures and Tables

**Figure 1 biomolecules-12-01035-f001:**
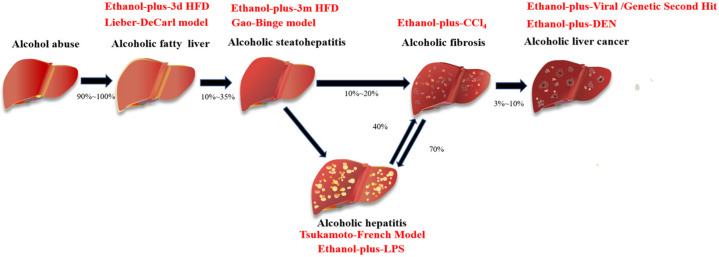
Commonly used experimental models for various stages of ALD. Ethanol-plus-3d HFD, ethanol-plus-three days high-fat diet; Ethanol-plus-3m HFD, ethanol-plus-three months high-fat diet; Ethanol-plus-CCl_4_, ethanol-plus-carbon tetrachloride; Ethanol-plus-DEN, ethanol-plus-N-Nitrosodiethylamine; Ethanol-plus-LPS, ethanol-plus-lipopolysaccharide.

**Figure 2 biomolecules-12-01035-f002:**
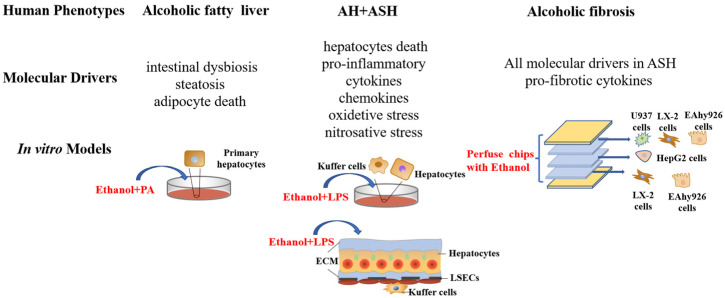
Commonly used in vitro models of ALD at different stages. ASH, alcoholic steatohepatitis; AH, alcoholic hepatitis; Ethanol + PA, ethanol plus palmitic acid; Ethanol + LPS, ethanol plus palmitic acid; ECM, extracellular matrix.

**Table 1 biomolecules-12-01035-t001:** Comparison of rodent models of alcohol-associated liver disease.

Rodent Models	Feeding Mode	Pathological Features	Advantages and Disadvantages	References
Model Using Alcohol Alone				
Lieber–DeCarli model	Chronic ethanol feeding (4–12 weeks)	ALT and AST levels were elevated, and it caused a certain degree of liver damage but rarely caused liver inflammation.	Easy to perform mild steatosis Short-term feeding with no mortality rate	[12,13,111,114]
Tsukamoto–French model	Intragastric infusion (2–3 months)	ALT levels were significantly elevated, and it led to severe steatosis, fibrosis, and cirrhosis with focal necrosis and immune cell infiltration.	Difficult to perform the requirement for intensive medical care Long-term feeding with a high mortality rate	[117,118,119,120,121,122,123,124]
The chronic plus binge model	LDE diet plus a single binge or multiple binges	ALT and AST levels were significantly elevated, it caused fatty liver and inflammation as well as neutrophil infiltration, and aged mice were more susceptible to liver damage and inflammation.	Easy to perform Short-term (10 d) feeding with no mortality rate Long-term feeding plus multiple binges with a high mortality rate	[11,101,115,125,126,127,128]
Second Hit Models				
HFD plus ethanol	3dHFD plus ethanol or 3mHFD plus ethanol	ALT and AST levels increased obviously, it induced acute hepatitis and injury, and it increased the infiltration of liver neutrophils and reduced liver macrophages.	This model needs a longer modelling period Simulates acute steatohepatitis in obese alcoholics	[18,129,130]
Iron carbonyl plus ethanol	Ethanol plus iron carbonyl (0.12% *w*/*v* for the first week, 0.25% *w*/*v* for the second week and beyond, for 16 weeks)	ALT and AST levels increased obviously, and it resulted in moderate to severe fatty liver as well as central necrosis and inflammation of the liver lobules, liver fibrosis, and even cirrhosis.	Time-consuming and costly Overcomes the animals’ natural aversion to ethanol Showed most of the disease course and liver pathology of ALD	[118]
LPS plus ethanol	Alcohol gavage (6 g/kg bw) plus LPS (10 mg/kg bw)	The levels of ALT and AST were significantly increased, increasing the degree of necrosis and hepatic neutrophil infiltration.	Easy to perform severe steatosis and inflammation	[16,131,132]
	LDE diet for 4 weeks, 5% ethanol *w*/*v* gavage plus intraperitoneal injection of LPS (2 mg/kg)	Liver sections also showed lipid droplet accumulation and enhanced liver damage, with distinct areas of necrosis and inflammatory cell infiltration. Moreover, the levels of ALT were significantly increased.		
CCl_4_ plus ethanol	LDE diet plus CCl_4_ injection (0.5 μL/kg, once every 3 days for 8 weeks)	This resulted in an exacerbation of hepatic fibrosis, characterized by increased activation of HSC.	Easy to perform liver fibrosis Toxic components Long-term model showed most of the disease course and liver pathology of ALD, but it was time-consuming	[17,133,134,135,136,137]
	Inhaling CCl_4_ plus ethanol for 7 wk (4% in the first week, 8% in the second week, and 16% afterwards)	It caused significant fibrosis within 4 weeks and strong proinflammatory reaction.		
	Intraperitoneal injection of CCl_4_ (0.2% mL/kg for 28 weeks) combined with LDE diet (containing 5% *v*/*v* ethanol for 10 weeks)	It caused hepatic steatosis, inflammation, fibrosis, hepatocyte swelling, and tumour nodules in mice.		
APAP plus ethanol	APAP (300 mg/kg bw) plus ethanol 4 g/kg every 12 h × 5 doses or three weekly ethanol binges	Significantly elevated ALT and AST levels, causing infiltration of erythrocytes in the space of Disse at 2 h after APAP treatment.	Displayed severe hepatotoxicity and early ALD features in the short term	[138]
DEN plus ethanol	LDE diet plus DEN (75 mg/kg for first three weeks and 100 mg/kg for the next three weeks)	Liver damage continued to increase and eventually showed increased recruitment of precancerous liver macrophages with mixed M1/M2 phenotype.	Reflected alcohol-induced HCC in terms of histology and genetics	[139,140]
	DEN10 (mg/kg) was injected intraperitoneally in 2-week-old mice and LDE diet (4.8% alcohol for 3–7 weeks) was given at 3 months of age	Alcohol intake significantly increased the number of surface tumours in mice.	Visible superficial tumours were induced and serum alpha-fetoprotein levels increased to 3 times the normal level	

**Table 2 biomolecules-12-01035-t002:** Comparison of models of alcohol-associated liver disease in vitro.

Models	Stimuli	Cell Strain	Related Indicators	Characteristic	References
2D monolayer cell culture model	Ethanol (100 mM, for 24 h)	HepG2 AML-12	Increased cellular TG levels as well as FASN and SREBP-1C expression Elevated CYP2E1 expression in HepG2 cells, resulting in increased GSH, ROS and superoxide in mitochondria	Easy to execute Can cause lipid accumulation and oxidative stress in cells	[178,179,180]
	Ethanol (100 mM, for 48 h) plus LPS (500 ng/mL, for 6 h)	RAW264.7 cells Peripheral blood monocytes (PBMCs)	TLR4 protein concentration Marked elevation in ROS production Increased release of inflammatory factors	Easy to perform Can cause cellular apoptosis and mitochondrial damage and autophagy response	[146,182,183]
	1 or 5 μg·mL^−1^ ethanol plus PA (400 μM, for 24 h)	Primary rat hepatocytes	Significantly increased expression of CHOP, ATF4, and XBP-1 in the nucleus and increased caspase-3 fragmentation in the cytoplasm	Increased lipid accumulation, endoplasmic reticulum stress, and caspase activation	[74]
3D cell culture model	Add ethanol (100 Mm) to the culture medium and culture for 7 days	Integrate hFLMCs into hEHOs	Increased activity of CYP2E1 and CYP3A4	Extracellular matrix deposition and apoptosis and oxidative stress	[191,192]
Liver-on-chip	Sterile filtered ethanol (60 mM, for 48 h)	HepG2, LX-2, EAhy926, and U937 cells	Measured biomarkers including Ve-cadherin, eNOS, VEGF, and α-SMA to understand the cell-to-cell communication between different types of hepatocytes during ALD	Increased the activity of liver cancer cells and maintained high liver function	[193,194]
	Perfuse the chip with ethanol plus LPS (133 mM for 48 h)	Primary hepatocytes, LSECs, and Kupffer cells	Significantly increased expression of proinflammatory cytokines interleukin-6 (IL-6) and TNF-α, and expression of MRP2 in large plaques	Intracellular accumulation of lipids, development of oxidative stress, and cholesterol synthesis dysregulation	[195,196]

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
