# Peer review of "Advancements in the Alcohol-Associated Liver Disease Model"

_biomolecules, 2022, doi:10.3390/biom12081035_

Round 1

Reviewer 1 Report

In this review paper, Zhu et al. summarized studies for in vivo and in vitro models to simulate alcoholic liver disease. The manuscript is written well and helps to understand recent advances in this area. However, there are several issues to be addressed as below.

1.     This paper distinguishes alcoholic hepatitis (AH) from alcoholic steatohepatitis (ASH). Please define differences among these diseases.

2.     Please spell out all abbreviations such as “iNKT” at the first appearance in the main text.

3.     Table 1 is hard to see because the layout is not good. Please consider to reconstruct the layout including a modification to reduce the columns.

4.     Lines 187-188 – splenomegaly and hyperbilirubinemia are not typical features of acute pancreatitis.

5.     Lines 198-199 – is the description “After 9 h of euthanasia was performed” correct? A blood test 9 h after euthanasia is atypical. Please check.

6.     There are some typos such as “singel” (line 342 and Table 2), “4.22. D cell culture model” (line 360), “4.33. D cell culture and liver-on-chip to model ALD” (line 375), “diabetes, diabetes” (line 420).

Reviewer 3 Report

This review summarizes the use of animal and in vitro models of alcoholic liver disease. The manuscript is poorly organized, has continuity and flows poorly. Many statements come from nowhere making it difficult to follow exactly what the authors are trying to convey. At times it seemed that statements from other manuscripts have been randomly inserted into the text and really don't fit. I found that the review of the models was cursory at best. For example, it is not clear from the description provided by the authors how the ethanol is administered in the binge model. Is it administered by gavage by injection or by some other method. Likewise, the cell culture models are poorly described the description of the 2-D models do not even method mentions ethanol. It's not clear to me what new information this review provides as other excellent reviews on this topic have been published. Many statements are made in the discussion that we're not thoroughly covered in the manuscript itself. Unfortunately, many of these statements are not referenced. The authors should emphasize any new information that they are providing.

Round 2

Reviewer 2 Report

the authors have been responsive to suggestion and revised the manuscript accordingly.

Reviewer 3 Report

I found that the authors response to my earlier comments were cursory at best. I does not appear that the authors understood that this manuscript required substancial rewriting, reorganization, and proofreading. This did not occurr. Review articles must be clear to be effective and useful. 

Response to comment 1. The authors state that they have made substantial revisions to the manuscript and revised the logic of the paper and yet the revised manuscript is organized exactly as the original submission. 

Response to comment 2. I still found the description of the models was cursory at best. The fact that the authors included that the binge in the NIAAA model was by garage was really the only apparent addition. The addition of the primate and zebrafish models was good but these sections as with most of the manuscript are poorly written. Additionally, the advantages and disadvantages of the use of each model was not clearly stated for each model. It would've been useful to include this in a summary at the end of the description of each model.

Response to comment 3. The descriptions of the in vitro models were slightly improved. It would've been useful if the authors would've included specific examples of the use of the 2-D models and the use of inducible pluripotent stem cells in the study of ALD.

Response to comment 4. I still find very little new information is provided in this manuscript. The author state that they describe pathological mechanisms of ALD. Again at the very best the mechanisms the authors describe are very superficial. For example how does acetaldehyde cause oxidative stress? How does CYP 2E1 cause oxidative stress? The author state that ablation of Kupffer cells in the liver ameliorates ALD but they do not state the proposed proposed mechanism behind this. Additionally, Kupffer cells are resident liver macrophages. They are not recruited to the liver. It is misleading statements like this that are unacceptable in a review article.

Response to comment 5. Although the discussion was expanded and slightly improved many bold statements were made that were not referenced. Review articles should be sources where one can find the source of these statements.

Summary I do not find this revised manuscript to be much improved. As I stated in the first review of this manuscript it should be clear what new information is being provided. All of the information should be provided in a clear, concise, logical, and straightforward manner. I did not find this to be the case in this manuscript.
